# Novel Roles of Nanog in Cancer Cells and Their Extracellular Vesicles

**DOI:** 10.3390/cells11233881

**Published:** 2022-12-01

**Authors:** Mikako Saito

**Affiliations:** Department of Biotechnology and Life Science, Tokyo University of Agriculture and Technology, Tokyo 184-8588, Japan; mikako@cc.tuat.ac.jp

**Keywords:** Nanog, melanoma, extracellular vesicles (EVs), cancer cell, cancer metastasis, immune cells, extracellular vesicle (EV)-based vaccines

## Abstract

The use of extracellular vesicle (EV)-based vaccines is a strategically promising way to prevent cancer metastasis. The effective roles of immune cell-derived EVs have been well understood in the literature. In the present paper, we focus on cancer cell-derived EVs to enforce, more thoroughly, the use of EV-based vaccines against unexpected malignant cells that might appear in poor prognostic patients. As a model of such a cancer cell with high malignancy, *Nanog*-overexpressing melanoma cell lines were developed. As expected, *Nanog* overexpression enhanced the metastatic potential of melanomas. Against our expectations, a fantastic finding was obtained that determined that EVs derived from *Nanog-*overexpressing melanomas exhibited a metastasis-suppressive effect. This is considered to be a novel role for Nanog in regulating the property of cancer cell-derived EVs. Stimulated by this result, the review of Nanog’s roles in various cancer cells and their EVs has been updated once again. Although there was no other case presenting a similar contribution by Nanog, only one case suggested that NANOG and SOX might be better prognosis markers in head and neck squamous cell carcinomas. This review clarifies the varieties of Nanog-dependent phenomena and the relevant signaling factors. The information summarized in this study is, thus, suggestive enough to generate novel ideas for the construction of an EV-based versatile vaccine platform against cancer metastasis.

## 1. Introduction

The development of effective vaccines to prevent cancer metastasis is a socially important and urgent issue [1,2]. Although the quantity of target cancer cells or cancerous cells in prognostic patients might be very small, they can produce metastasis, as well as reactivate primary tumor sites, by the self-seeding of circulating tumor cells [3]. For such cases, the use of vaccines is well understood to be a strategically promising method. Immune cells can respond to malignant cells and activate protection systems that destroy or render them harmless. In fact, immune cells, such as dendritic cells, have been recognized as efficient resources for extracellular-vesicle (EV) vaccines. The only idea yet to be considered in the research, however, is whether immune cells can appropriately respond to cells with a high degree of malignancy. Cancer cells remaining in prognostic patients are likely to be cells with a high resistance to drugs and chemical stress [4,5]. Those malignant cells and EVs should, therefore, be crucial targets that might conversely undermine immune-protection systems. Therefore, a great expectation has arisen in the field for a novel idea to convert negative malignant factors to positive ones that support immune functions. The potential roles of cancer cell-derived EVs, as well as immune cell-derived EVs, should be considered for the construction of EV-based versatile vaccine platforms against cancer metastasis.

## 2. Why Nanog?

The first step of our experiment was to create a malignant cancer cell line with high metastatic potential. Mouse melanoma cell lines, B16-F10 and B16-BL6, were selected as the baseline for developing a novel cell library. These cell lines were genetically modified to create *Nanog*-overexpressing cell lines *Nanog*^+^F10 and *Nanog*^+^BL6. Nanog is a principal factor essential for the maintenance of the undifferentiated state (pluripotency, stemness) of embryonic stem cells. Nanog was thought to be able to increase the stemness of various other cells, and the stemness of cancer cells was suggested to be a crucial factor of malignancy. As expected, *Nanog* overexpression could enhance the metastatic potential of melanomas, indicating that a melanoma was made more malignant. EVs derived from B16-F10 cells exhibited a metastasis-promotion effect in the same way as those reported in other studies about other cancer cell-derived EVs. Unexpectedly, however, EVs derived from *Nanog*^+^F10 cells exhibited a metastasis-suppression effect (Figure 1). Such a Nanog-dependent effect of EVs was also observed for colon cancer. This result began with a simple idea of *Nanog* overexpression to increase stemness. However, it revealed an attractive phenomenon from the perspective of EV-based vaccines. Therefore, it would be important to investigate the detailed role of Nanog in this phenomenon.

## 3. Roles of Nanog in Cancer Cells

A high quantity of research papers have reported the potential roles of Nanog in various types of cancers. There were two or three recently published papers for each type of cancer selected, and they are summarized in Table 1. Short comments for respective papers are described, below, under Section 3.1, Section 3.2, Section 3.3, Section 3.4, Section 3.5, Section 3.6, Section 3.7, Section 3.8, Section 3.9, Section 3.10, Section 3.11, Section 3.12 and Section 3.13.

### 3.1. Breast Cancer

The overexpression of *NANOG* increased cell adhesion. Additionally, *p53*, a tumor-suppressive gene, decreased. Concomitantly, the expression of downstream factors, such as *Gadd45a*, also decreased. The enhancement of *Nanog* expression promoted migration and invasion activities. Tumorigenesis was not induced by *Nanog* overexpression alone but by the co-expression with *Wnt-1* [6].

Treatment with mTOR inhibitors and chemotherapeutic agents increased *NANOG* expression in a similar manner to hypoxia. Concurrently, the translation of a subset of *SNAIL* and *NODAL* mRNA isoforms was activated. The accumulation of these proteins enhanced the stem cell phenotype, increased drug resistance, and promoted metastasis [7].

### 3.2. Cervical Cancer

CD59 binds to C8 and C9 and, therefore, inhibits the formation of the membrane attack complex (MAC) that requires C9. Therefore, complement-dependent cytotoxicity (CDC) via MAC is inhibited by CD59. In a *NANOG*-overexpressing cell line (CaSki-*NANOG*), *CD59* was up-regulated, and the resistance to CDC increased. NANOG directly bound to the *CD59* promoter to enhance its expression activity [8].

According to the cancer immunoediting theory, heterologous tumor cells are continuously subject to host immune surveillance [32,33]. Cells vulnerable to immune surveillance are eliminated, while cells that evade detection and killing proliferate. Based on this idea, a method to create cancer cells with high immune-resistance levels, the vaccination-induced cancer evolution (VICE) method, was developed. TC-1(P3), which was obtained by repeating the subculturing process (removing the cancer cells that were inoculated into mice and then inoculating them into mice again) by this method three times, was the first cell line. *Nanog* expression was increased 10-fold compared to TC-1(P0). An increase in stemness markers (*CD133*, *CD44*, *aldehyde dehydrogenase* (*ALDH*)) was also observed [9]. It was shown that the higher the degree of malignancy of cancer cells, the higher the expression level of *Nanog*.

### 3.3. Colon Cancer/Colorectal Cancer

In colon cancer, LGR5 and NANOG are assumed to be stem cell markers. Therefore, the possibility of therapeutics targeting these markers was investigated in this study. As an example, furin, which belongs to the subtilisin-like proprotein convertase family, was investigated. Furin is involved in the activation of the functions, such as calcium transport, in colon cancer. Inhibitors of furin, such as PDX-1, Spn4A, and decanoyl-RVKR-chloromethylketone (CMK), were applied to investigate the effect of furin inhibition in vivo. As a result, it was understood that furin inhibition reduced the expression of stem cell markers and the malignancy of cancer cells [10].

In colorectal cancer, serum deprivation induced increased chemoresistance and enhanced dormancy through the increased expression of dormancy markers, and it also induced enhanced *Nanog* expression. The knockdown of *Nanog* abolished dormancy, whereas the overexpression of *Nanog* promoted dormancy through the transcription of *P21* and *P27*. In the dormant state, cancer cells are malignant. Thus, enhanced *Nanog* expression is a factor in malignant transformation [11].

### 3.4. Embryonic Carcinoma

NANOG was shown to promote tumorigenesis in embryonic carcinomas. miRNAs that suppress *NANOG* expression were sought. The upstream factors of *NANOG* were surveyed. PKC was confirmed to be involved in the regulation of *NANOG* expression. A genome-wide analysis of miRNA expression was performed in the embryonal carcinoma cell line NT2/D1 in the presence of the *PKC* activator phorbol 12-myristate 13-acetate (PMA). As a result, an increased expression of *MIR630* was confirmed. The transfection of *MIR630* into embryonic carcinomas suppressed *NANOG*. The reactive site was *NANOG 3′UTR* [12].

### 3.5. Somatic Cancer

HeLa (cervical cancer) and HCT116 (human colon cancer) were used as somatic cancer cells. Rad51 is a protein involved in the homologous recombination (HR) repair of DNA damage. This protein prevents cancer cells that have been damaged by chemo or radiation therapies from dying. Therefore, Rad51 inhibitors were considered as effective for cancer treatment. Nanog was shown to be effective as a Rad51 inhibitor. Nanog interacted with Rad51 at the C or CD2 domain. Nanog-C/CD2 peptides were directly delivered to somatic cancer cells via nanoparticles or cell-membrane permeable peptides. The introduction of Nanog or moieties contributed to tumor suppression [13].

### 3.6. Hepatocellular Cancer

*NANOG* was activated by the *TLR4-E2F1* pathway. *NANOG* suppressed mitochondrial *oxidative phosphorylation* genes (*OXPHOS*) and enhanced *fatty acid oxidation* (*FAO*) in tumor-initiating stem-like cells. *FAO* enhanced self-renewal and chemoresistance properties. On the other hand, restoring *OXPHOS* suppressed the self-renewal property [15].

### 3.7. Melanoma

The relationship between different motility modes and metastatic potential in human melanoma A375 was also investigated in this study. Mobility includes mesenchymal and amoeboid migrations [34]. A375 showed a mesenchymal motility mode, but the overexpression of *NANOG* or *OCT4* increased amoeboid migration, resulting in an increased metastatic potential [17].

*Nanog* was up-regulated in mouse melanomas under hypoxia. This increased regulatory T cells (Treg) through the increased expression of *Tgf-β1*. Tregs are *CD4*^+^T cells that release the anti-inflammatory cytokine IL-10 and suppress immune responses. As a result, the proliferation and metastasis of cancer cells were promoted. The targeted inhibition of Nanog reduced Treg-like immunosuppressive cells and increased *CD8*^+^T cells (cytotoxic T cells), resulting in the suppression of cancer growth and metastasis [18].

### 3.8. Ovarian Cancer

Hexokinase 2 (HK2) is one of four isoenzymes. *HK2* was overexpressed in ovarian cancer and showed significantly higher expression levels in ascites and metastases. Cell migration and invasion were enhanced by a *NANOG*-non-mediated pathway, *HK2* ⇒ *FAK* ⇒ *ERK1/2* ⇒ *MMP9*, and stem cell properties were enhanced by a *NANOG*-mediated pathway, *HK2* ⇒ *FAK* ⇒ *ERK1/2* ⇒ *NANOG*, *SOX9* [21].

### 3.9. Pancreatic Cancer

In rare and highly malignant cancer stem cells, the *hedgehog/glioma-associated oncogene homolog* (*HH/GLI*)-signaling pathway regulates self-renewal, initiates and sustains tumor growth, and promotes drug resistance and metastasis [35]. The inhibitory effect of natural α-mangostin on this signaling pathway was examined. As a result, the expression of target genes (*Nanog*, *Oct4*, *c-Myc*, *Sox-2*, and *KLF4*) of this signal transduction system was inhibited, and an antitumor effect was observed. Conversely, the overexpression of *Nanog* abolished its inhibitory effect, suggesting that the effect of α-mangostin was mainly obtained by inhibiting *Nanog* expression. At the same time, it was concluded that the method targeting *Nanog* is preclinically effective for the prevention and treatment of pancreatic cancer [22].

### 3.10. Prostate Cancer

The relationship between cell–cell adhesion and the malignancy of prostate cancer cells DU145, PC3, and 22Rv1 was investigated. The overexpression of *NANOG* enhanced the ability to evade attacks from the NK cell MTA cell line (CD4 and CD56-positive T-cell line) [23]. NANOG suppressed the expression of ICAM1, a cell-adhesion molecule. Without ICAM1 on the cell surface, NK cells cannot recognize it, and cancer cells escape attack from NK cells.

### 3.11. Squamous Cell Carcinoma

In esophageal squamous cell carcinomas (ESCCs), the knockdown of *NANOG* clearly reduced cancer cell proliferation and the ability to resist drugs. It was presumed that *IL-6/STAT3* was down-regulated [25].

In the case of head and neck squamous cell carcinoma (HNSCC) cells, a comparative analysis of CD44^+^ cells (indicator of stemness) and control CD44^(−)^ cells revealed that *Nanog* or *ERK1/2* was highly expressed in CD44^+^ cells. Thus, it was determined that they exhibited migration ↑, invasion ↑, radiotherapy resistance ↑, and EMT ↑ properties [26]. *Nanog* and *ERK1/2* appeared to exhibit synergistic effects.

For HNSCC, 348 postoperative patients were also investigated [27]. As a result, NANOG protein was highly expressed in 72%, and SOX2 was highly expressed in 30%. The prognosis was better in NANOG’s and SOX2′s high expressions. In other words, NANOG and SOX2 can be used as good prognostic markers. Moreover, NANOG was also tumor site-specific and correlated with a favorable prognosis for pharyngeal tumors (rather than laryngeal) [27]. NANOG is probably uniquely considered a good prognostic marker. NANOG also serves as an independent prognostic factor in nasopharyngeal carcinomas [28].

The case of oral squamous cell carcinomas (OSCCs) was also investigated in 120 patient samples following surgery. As a result, the expressions of *NANOG* and *OCT4* were higher in patients with lymph node metastases than in those without lymph node metastases, suggesting the possibility of NANOG as a malignant prognostic marker. However, protein and mRNA expression levels sometimes did not match. At the mRNA level, it was positively correlated with other cancer malignancy-associated factors: *OCT4*, *SOX2*, *NOTCH1*, *AGR2*, and *KLF4* [29].

### 3.12. Cancer Stem Cells

The validity of NANOG as a cancer stem cell marker was discussed. It was proposed that NANOG might be considered as one of the markers, based on the following observations of multiple types of cancer cells, when *NANOG* is overexpressed. Following the enhancement of *NANOG* expression, there appeared increased expressions of *BMI* and *SNAIL1/2*, followed by the suppression of *E-cadherin* expression in various cancer cells (glioblastoma, non-small lung cancer, HNSC, colon cancer, and A549). In addition, an increased expression of *NANOG* ⇒ *STAT3* ⇒ *miR21* was followed by the down-regulation of *programmed cell death 4* (*PDCD4*), resulting in the enhanced anti-apoptotic and chemoresistance properties of cancer cells. All of these are factors that increase the migratory ability, proliferative ability, and epithelial–mesenchymal transition (EMT), resulting in an increased malignancy as cancer cells [30].

On the other hand, the hedgehog signaling factor binds to PATCH1/2 (which is the receptor) and abolishes the inhibitory effect of PATCH1/2 on SMO. As a result, *SMO* ↑ ⇒ *GLI1* ↑ ⇒ GLI1 nuclear translocation ⇒ *NANOG* promoter-activation ⇒ *NANOG* mRNA ↑ ⇒ NANOG protein ↑ ⇒ *GLI1* ↑ (positive feedback). On the other hand, *p53* (which induces apoptosis) represses the *NANOG* promoter ⇒ *NANOG* mRNA ↓ ⇒ NANOG protein ↓. Regardless of *NANOG* expression levels, *NANOG* represses *p53*, creating negative feedback. These positive and negative feedbacks suppress apoptosis, maintain cancer stemness, and contribute to cancer malignancy [30].

### 3.13. PD-1-Treated Patients and Their Model Mice

Programmed cell death protein 1 (PD-1) inhibitors and PD-L1 inhibitors are a group of checkpoint-blocking anticancer agents that block the activity of PD-1 and PDL1 immune-checkpoint proteins present on the cell’s surface. This immune-checkpoint inhibitor has emerged as a frontline therapy for several types of cancer. Using the transcriptional data obtained from cancer patients treated with PD-1 therapy and a newly established murine preclinical anti-PD-1 therapy-refractory model, NANOG was identified as a factor that enhanced patients’ resistance to immune-checkpoint inhibitors. NANOG regulated this immune checkpoint by suppressing T-cell infiltration and increasing resistance to killing by cytotoxic T lymphocytes (CTLs) through a histone deacetylase 1-dependent (HDAC1-dependent) regulation of *CXCL10* and *MCL1* [31].

### 3.14. Summary of NANOG Roles

The role of Nanog, which has been clarified for various cancer cells, is related to the growth and migration of cancer cells themselves, as well as the interaction with various extracellular factors from the perspective of the effects on cancer cell functions (Figure 2). The following points summarize the contents of Table 1.

(a)High levels of *Nanog* expression are associated with increased malignancy, which has been observed in many types of cancers. The only exception is the case of HNSC.(b)Nanog targeting, alone, does not necessarily lead to cancer cytocide.(c)The degree of malignancy of cancer cells is not solely governed by *Nanog*.(d)Cancer cells with high levels of *Nanog* expression have high metastatic potential. It shows potential as a marker of malignant prognosis. Indeed, Nanog has shown promise as a marker for predicting the efficacy of PD-1 therapy.(e)Molecular mechanisms, leading to malignant transformations, greatly differ depending on the type of cancer. There are almost no research reports about why *NANOG* signaling differs depending on cancer types. This point should be clarified for the use of Nanog as a therapeutic target.(f)From a therapeutic perspective, the enhancement of immune functions is essential and, therefore, novel ideas are required to combine Nanog-targeting therapy with immunotherapy.

## 4. Nanog Overexpression in Melanoma

### 4.1. Transcriptome Analysis

The transcriptome analysis of mouse melanoma cells was conducted to clarify the differential expression intensities between a cell line, B16-BL6, and its *Nanog*-overexpressing cell line *Nanog*^+^BL6. The up-regulated top-16 genes and down-regulated top15 genes were depicted [19]. The functional roles of up-regulated top-7 genes and down-regulated top-3 genes are illustrated in Figure 3A,B, respectively.

Slc37a4 is a protein that transports glucose-6-phosphate (G6P) from the cytosol to the endoplasmic reticulum (ER). G6P is dephosphorylated in ER and released as glucose out to intercellular space or blood vessels. When cancer cells form colonies, glucose diffusion from the outer solution to the central cells takes a much longer time when compared to cells on the outer surface of the colony. In such a case, if a series of cells in contact with each other can relay glucose, glucose transport can be performed rapidly. The increased expression of *Slc37a4* may contribute to the activation of such a glucose relay. The accelerated glucose supply throughout the colony will accelerate cell growth. The acceleration of energy production as ATP is facilitated by five genes (*mt-Co2*, *mt-Atp8*, *mt-Atp6*, *mt Co3*, and *mt-Nd4*) that may contribute to the acceleration of oxidative phosphorylation. *Vesicle-associated membrane protein 8* (*Vamp8*) is involved in surviving the emergency of starvation. When cancer cells are placed in a state of starvation, they transport their own cytoplasm into autophagosomes, digest it, and use the nutrients.

The most down-regulated gene is *Jak*. Immunosuppression and malignant tumors are caused by the dysfunction of the *Jak-STAT*-signaling pathway. The down-regulation of *Jak* causes a similar condition and also produces more malignant melanomas. Glut4 facilitates glucose uptake. Tbc1d1 suppresses this uptake of glucose. Therefore, the suppression of *Tbc1d1* stimulates glucose uptake activity and promotes cancer cell growth. Regarding *Tgf-β1*, however, it is necessary to consider its dual roles: tumor-suppressive in early stage tumors but tumor-promotive in advanced cancers [36,37,38]. Tumor-promotive roles are the promotion of angiogenesis, immunosuppression, and apoptosis induction. Transcriptome analysis indicated the down-regulation of *Tgf-β1*, although melanoma cells were made more malignant, which was supported by in vitro and in vivo tests.

### 4.2. Experimental Analyses

The in vitro and in vivo tests were conducted to investigate the effects of *Nanog* overexpression on the functional roles of melanoma cells. The cell lines were B16-F10, B16-BL6, *Nanog*^+^F10, and *Nanog*^+^BL6. The characteristic functions to be studied for a metastatic property evaluation are summarized in Figure 4.

The glucose uptake activity is greater in cancer cells than in normal cells, and an analytical method used for visualizing glucose uptake activity has been introduced into cancer diagnostic methods. A pathological observation method that utilizes fluorescent glucose (Figure 5) was developed to distinguish between cancer cells, normal cells, and cells likely to become cancerous [39,40]. Normal cells only take up 2NBDG (D-type fluorophore), whereas cancer cells take up both 2NBDG and 2NBDLG (L-type fluorophore). In fact, it was confirmed that four melanoma cell lines tested could take in both 2NBDG and 2NBDLG. Furthermore, it was suggested that the total uptake of 2NBDG and 2NBDLG might be used as a marker of the degree of cancer cell malignancy.

Cell–cell glucose relay was suggested to be one of Slc37a4′s roles. Accordingly, it suggested the promotion of glucose uptake as well. The knockdown of *Slc37a4* caused a decrease in the uptake rate of 2NBDG, but there was no effect on the uptake rate of 2NBDLG [unpublished data]. Therefore, the up-regulation of *Slc37a4* contributed to the increase in glucose uptake.

Since the expression level of *Tgf-β1* was a controversial matter, it was analyzed at mRNA and protein levels. As a result, its expression was down-regulated at both levels [19]. Another study conducted elsewhere [18] demonstrated a conflicting result. *Nanog* expression in melanomas increased under a hypoxic condition, and the increase in *Tgf-β1* expression followed. We suspected that this inconsistency might be caused by the difference in the expression level of *Nanog*. The *Nanog* expression level, up-regulated by a hypoxic condition, might be much lower than that up-regulated by genetic overexpression. The response of *Tgf-β1* was concluded to be *Nanog*-expression level-dependent.

The expression of *matrix metalloproteinase* (*MMP*)s was studied since MMPs were believed to be relevant to invasion, although they were not included in the top 31 genes. They observed the increase in *MMP9*, a secretion-type MMP [19].

The interaction with immune cells was thought to principally occur via the EVs described below. The involvement of macrophage and Treg was investigated.

The results of the studies performed show that *Nanog* overexpression made melanoma cells more aggressive. In addition, melanoma cell lines with co-overexpressing *Nanog*, with *Oct3/4* and/or *Sox2*, were created in order to further enhance stemness. Consequently, however, any combination could not create a cell line with a greater metastatic potential than the cell line with the overexpression of *Nanog* alone.

## 5. Properties of EVs Secreted from Cancer Cells

### 5.1. Tumor-Promoting Effect

The functional roles of dendritic cell-derived EVs and cancer cell-derived EVs have been well discussed in the literature [1,41,42]. Dendritic cell-derived EVs are regarded as promising materials for immunotherapy. In contrast, cancer cell-derived EVs are considered unsuitable. In fact, all of the 31 cases summarized in [42] showed that the effect of cancer cell-derived EVs on immune cells was the suppression or inactivation of immune activity. The active substances delivered by EVs were unknown in 10 out of 31 cases. Other cases, however, were six Fas cases, six TGF-β cases, and five miRNA cases. In the case of melanomas involving the Fas ligand, Jurkat and other lymphoid cells induced apoptosis associated with caspase activation [43]. Colon cancer also expressed the Fas ligand and TNF-α at the same time, resulting in the induction of T-cell apoptosis both in vivo and in vitro.

When cancer cell-derived EVs are taken up by other cancer cells of the same type, they change the properties of those cancer cells. There were eighteen types of cases summarized, and in all cases, EVs increased the proliferation, migration, invasion, EMT, and metastasis of cancer cells that took them in [4]. It also promoted the polarization of macrophages to the M2 type (tumor-promotive). In these examples, EV-delivered active substances included integrin α_V_β_6_, apolipoprotein E, EGFR, Wnt4, IL-6, and TGF-β, as well as cell-specific miRNAs and long non-coding (lnc) RNAs. On the other hand, cancer cell-derived EVs are also transported to normal fibroblasts, and once taken up, the fibroblasts release EVs that have suppressive effects on immune cells [44].

Furthermore, cancer cells that have undergone anticancer drug treatment may be highly resistant to the drug. EVs released from such highly resistant cancer cells may change the non-resistant allogeneic cells to resistant cells. The effects of EVs derived from cancer cells that were resistant to anticancer agents, such as tamoxifen [45,46], cisplatin [47,48], and gemcitabine [49], were investigated. Cancer cells exposed to those EVs became more resistant to respective anticancer agents. EVs secreted from liver cancer stem cells induced *Nanog* in differentiated cancer cells, resulting in increased resistance to the anticancer drug regorafenib [50]. Small EVs secreted from gastric cancer cells enhanced the stemness of other gastric cancer cells and increased their resistance to oxaliplatin [51].

In another case, temozolomide (TMZ)-resistant and sensitive tumor cells were obtained from each of the TMZ-resistant (n = 36) and sensitive (n = 33) glioma patients. *Circular RNA circ_0072083* expression was increased in resistant cells, and its knockdown reduced resistance, concomitantly reducing *NANOG* expression. EVs containing *circ_0072083* released from resistant cells increased the resistance of sensitive cells to TMZ both in vitro and in xenograft models [5].

### 5.2. Metastasis-Inhibitory Effect

Cancer cells that have undergone chemotherapy, radiation therapy, and heat stimulation may increase resistance to each factor. This creates an increase in malignancy. EVs released from such malignant cancer cells enter other cancer cells and strengthen their resistance to the same factor as described above. However, EVs are also taken up by immune cells, such as dendritic cells. As a result, dendritic cells receive malignant cancer cell information and damage-related molecular patterns (DAMPs), such as DNA and RNA, which may enhance antitumor activity by activating intracellular virus-sensing pathways and producing inflammatory cytokines.

EVs released from breast cancer cells treated with the anticancer drug topotecan contained DNA that activated dendritic cells via a *stimulator of interferon gene* (*STING*) signaling [52]. In addition, when breast cancer model cells were irradiated with therapeutic radiation, the EVs released from these breast cancer cells were taken up by dendritic cells, and then, they activated the *cyclic GMP-AMP synthase* (*cGAS*) within the dendritic cells. In this case, dendritic cells were also activated via *STING* signals. In vivo, the EVs elicited a *CD8*^+^T-cell response and presented tumor-preventive effects [53].

The final case was the metastasis-suppressive effect of melanoma-derived EVs that initiated this review. There are still a few cases of metastasis inhibitory effects by cancer cell-derived EVs.

## 6. Suppression of Cancer Metastasis by Melanoma-Derived EV

### 6.1. Comparison of Metastatic Potential between B16-F10 and Nanog^+^F10

Metastatic colonies were analyzed two weeks after the introduction of mouse melanoma B16-F10 and *Nanog*^+^F10 from the mouse tail vein. Preliminary studies revealed that the highest number of metastatic colonies was generated on the liver. Therefore, liver was focused on as a predominant target organ, and the number and volume of metastatic colonies were quantitatively analyzed. As a result, those of *Nanog*^+^F10 increased 2.5 and 2.4 times, respectively, compared to B16-F10 [20]. At the same time, in vitro tests were conducted separately to investigate cell proliferation and migration. The results support the enhancement of the metastatic potential of *Nanog*^+^F10.

### 6.2. Comparison of the Effect of B16-F10-EV and Nanog^+^F10-EV on Melanoma Metastasis

EVs released from B16-F10 and *Nanog*^+^F10 were obtained, and their vaccine effects were investigated. EVs (5 μg/100 μL PBS) or PBS (100 μL as a control) were injected into the tail vein of 5–6-week-old mice 3 times per week for 3 weeks, and subsequently, melanoma cells (5 × 10^5^ cells/250 μL PBS) were injected into the tail vein. Livers were separated and metastatic colonies were analyzed 2 weeks later. As a result, F10-EV increased metastasis, but *Nanog*^+^F10-EV suppressed metastasis [20].

### 6.3. Role of Tgf-β1 in the Anti-Metastasis Effect

As a mechanism of the metastasis-suppressive effect, (i) the effect of Tgf-β1 in EV and (ii) the tumor suppressive effect by immune cells (macrophages, Tregs) were presumed. Regarding (i), B16-F10, *Nanog*^+^F10, F10-EV, *Nanog*^+^F10-EV, and *Tgf-β1* knockdown cell lines and EVs obtained from them (*Tgf-β1*^(−)^-F10, *Tgf-β1*^(−)^-F10-EV) were used. *Tgf-β1* gene expression level and protein concentration (pg/μg EVs) in those cell lines—and EVs therefrom—were analyzed. The concentration of Tgf-β1 protein in EVs and the effect on metastasis were summarized as F10-EV (3.9 pg/μg EVs, promotive), *Nanog*^+^F10-EV (1.6 pg/μg EVs, suppressive), and *Tgf-β1*^(−)^-F10-EV (0.5 pg /μg EVs, suppressive) [20]. At high concentrations of Tgf-β1, the role of Tgf-β1 was the promotion of metastasis. In contrast, it turned into the suppression of metastasis at lower levels than a threshold of 1.6–3.9 pg/μg EVs.

Although there are few papers that report the quantitative studies conducted on the role of TGF-β1 in EVs, we obtained a couple of papers that may support the validity of such a threshold level. Exosomes derived from melanoma A375 cells contained 10–15 pg/μg TGF-β and inactivated T cells, suggesting a metastasis-promotive role [54]. In contrast, EVs derived from murine colon carcinoma cells that had been genetically modified with an overexpression of shRNA for *Tgf-β1* could induce tumor growth inhibition [55]. This suggests a metastasis-suppressive role at a sufficiently low level of Tgf-β1.

Tgf-β1 is involved in the regulation of EMT, suppressing EMT in the early stages of tumors but conversely promoting EMT in the late stages of tumors, but its concentration dependence is unclear [37,56,57]. Considering that Tgf-β1 is associated with various factors, it is conceivable that the concentration dependence of Tgf-β1 is not simple. Although the concentration dependence of Tgf-β1 revealed in this study is a phenomenon in a limited concentration range, it is highly suggestive in considering the multifaceted role of Tgf-β1.

### 6.4. Role of Immune Cells in Preventing Metastasis

Regarding (ii), we first examined the involvement of macrophages according to the test schedule. As a result of examining the expressions of six types of macrophage markers (pan-macrophage [*CD68*, *F4/80*], M1-type [*CD80*, *CD86*], and M2-type [*CD163*, *CD206*] macrophage markers), it was revealed that only the expression of the tumor-promotive M2-type marker *CD163* was significantly reduced [20].

Tumor-associated macrophages that exhibit tumor-promotive effects are M2-type macrophages, the majority of which are *CD163*-positive macrophages [58]. In addition, a positive correlation between the infiltration of *CD163*-positive macrophages into cancer and *PD-L1* expression in cancer has been reported from observing tissues of various cancer patients [59,60,61,62]. PD-L1 is an immunosuppressive receptor that suppresses T-cell proliferation and cytokine secretion [62]. Therefore, it is possible that the reduction in *CD163* by *Nanog*^+^F10-EVs reduced the suppressive effect on T cells, resulting in increased anti-tumor immunity activity.

Regarding (ii), we examined the effects of *Nanog*^+^F10-EV on *Foxp3*, which was a specific marker of Treg activation in the spleen, and observed that the expression of *Foxa3* was significantly suppressed. Treg inhibits cytotoxic T cells and macrophages by secreting cytokines, such as IL-10 and IL-35, and the cytotoxic T-lymphocyte antigen 4 (CTLA-4) ligand. Treg also inhibits acquired immunity by suppressing dendritic cells [63]. An artificial Treg inhibitor introduced into mice increased the tumor infiltration of cytotoxic T cells and suppressed subcutaneous melanoma cell tumors [64]. Therefore, it was inferred that the suppression of *Foxp3* in the spleen contributed to the metastasis-suppression effect of *Nanog*^+^F10-EV. 

### 6.5. Quantitative Analysis of the Effects of EVs Taken Up by Macrophages

The involvement of *CD163* was investigated by in vitro experiments using a macrophage cell line J774.1. In a similar manner to the in vivo test described above, *Nanog*^+^F10-EV caused a suppression effect on *CD163* expression in J774.1.

Subsequently, this suppressive effect of *Nanog*^+^F10-EV is further analyzed quantitatively. J774.1 cells are fractionated with a cell sorter according to the differences in EV uptake (Figure 6). Then, each fraction was tested for its invasion ability with Transwell test kits. The number of filtrated cells is counted and compared to the control. Higher uptake of *Nanog*^+^F10-EV will result in higher infiltration.

### 6.6. A Mechanism of Metastasis-Suppressive Effects by Nanog^+^F10-EV

Figure 7 summarizes a mechanism in which *Tgf-β1*, *CD163*, and *Foxp3* are involved. This is specific to melanomas.

## 7. Prospects for EV Cancer Vaccines

There are many studies on cancer cell-derived EVs. However, there are only a few papers [20,52,53] reporting metastasis-suppression effects. Among them, only one paper [20] addresses the *Nanog*-dependent phenomenon. Therefore, metastasis suppression by cancer cell-derived EVs is, to date, an extremely rare phenomenon. Recently, however, a similar phenomenon was observed for colon cancer-derived EVs. We expect that similar anti-metastasis effects will be observed for EVs derived from *Nanog*-overexpressing cells of other cancers in the near future.

To elucidate the molecular mechanism of the metastasis-suppression phenomenon, much effort must be focused on the quantitative analyses of various cargos of EVs. In the case of melanomas, Tgf-β1 was selected as a predominant factor, and an idea for its quantity threshold in EVs could be proposed. However, various other components coexist in EVs. It is necessary to analyze them to evaluate their possible involvement. Based on these analyses, we will be able to discuss whether metastasis is suppressed or promoted as a total effect.

*Nanog*^+^F10-EV and F10-EV are a suitable pair for the differential analysis of EV cargo components. Our plan is to analyze those components, such as miRNAs and cytokines. Although the analytical results only concentrate on melanoma-relevant matter, they are sure to contribute to the construction of an EV-based versatile vaccine platform against cancer metastasis.

## Figures and Tables

**Figure 1 cells-11-03881-f001:**
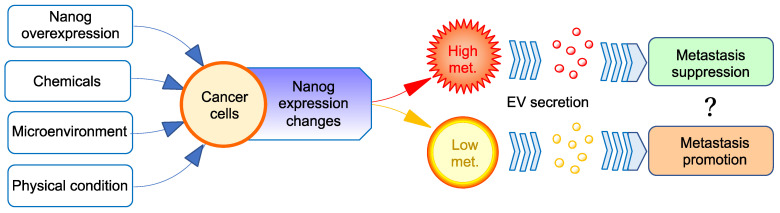
Paradoxical effects of EVs from *Nanog*-overexpressing cancer cells on their metastasis. *Nanog* expression level in cancer cells can be changed by genetic, chemical, microenvironmental, or physical factors. The higher the *Nanog* expression level, the higher the metastatic potential. The role of EVs in cancer metastasis has been thought to follow the metastatic potential of cancer cells, that is, EVs derived from metastatic cancer cells exhibit metastasis-promoting effects. However, in the case of cancer cells with a very high metastatic potential, contrary to our expectations, EVs may promote cancer metastasis.

**Figure 2 cells-11-03881-f002:**
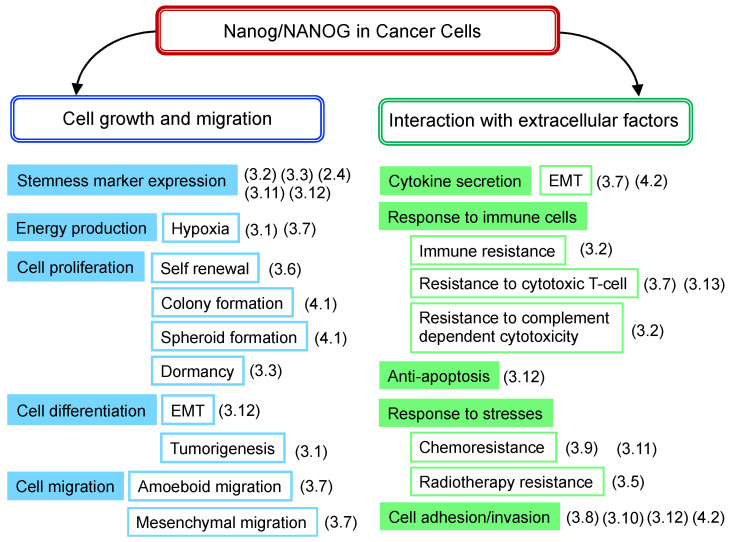
Nanog-dependent phenomena observed in cancer cells. Contents listed in Table 1 are rearranged according to the phenomena of cancer cells. (3.1), (3.2), denote the number of sub-sections with a description of the phenomenon.

**Figure 3 cells-11-03881-f003:**
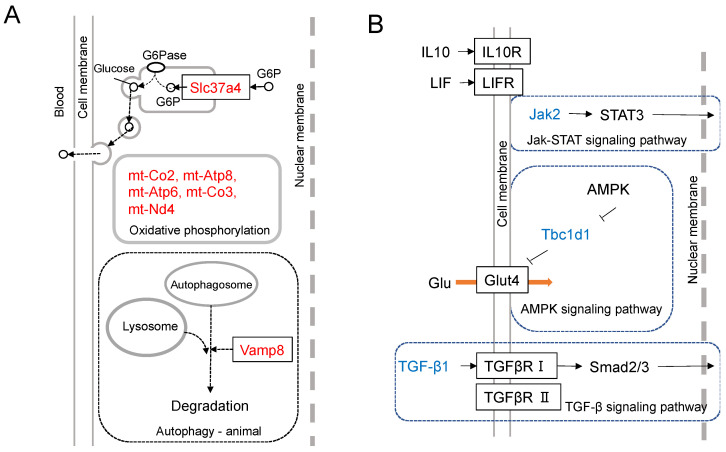
Results of transcriptome analysis. (**A**) Differential expression of *Nanog*-dependent genes between *Nanog*^+^BL6 and B16-BL6 cell lines. Assumed-role diagrams of up-regulated top-7 genes (*Slc37a4*, *mt-Atp6*, *mt-Nd3*, *mt-Co3*, *mt-Atp8*, *mt-Co2*, and *Vamp8*). (**B**) Differential expression of *Nanog*-dependent genes between *Nanog*^+^BL6 and B16-BL6 cell lines. Assumed-role diagrams of down-regulated top-3 genes (*Jak2*, *Tbc1d1*, and *TGF-β1*).

**Figure 4 cells-11-03881-f004:**
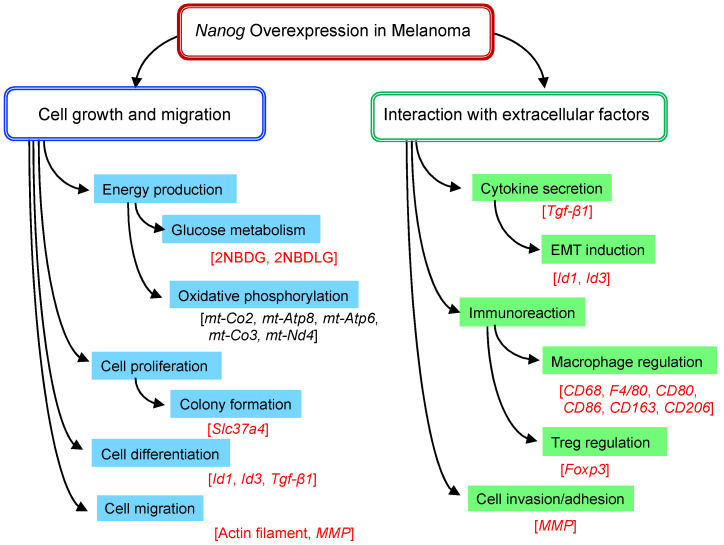
Nanog-dependent phenomena observed in melanoma. Relevant genes, proteins, or probes are described in brackets. Red letters in brackets indicate those analyzed by experiments; black letters in brackets indicate those analyzed only by transcriptome analysis.

**Figure 5 cells-11-03881-f005:**
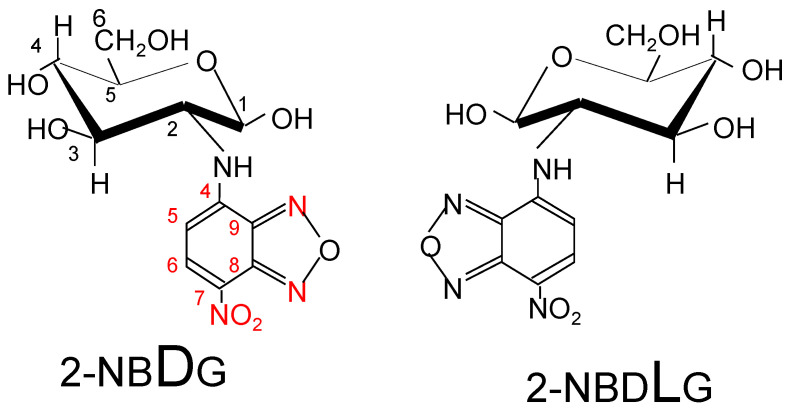
Fluorescent glucose analogs: 2NBDG: 2-[*N*-(7-nitrobenz-2-oxa-1,3-diazol-4-yl)amino]-2-deoxy-D-glucose, 2NBDLG: 2-[*N*-(7-nitrobenz-2-oxa-1,3-diazol-4-yl)amino]-2-deoxy-L-glucose.

**Figure 6 cells-11-03881-f006:**
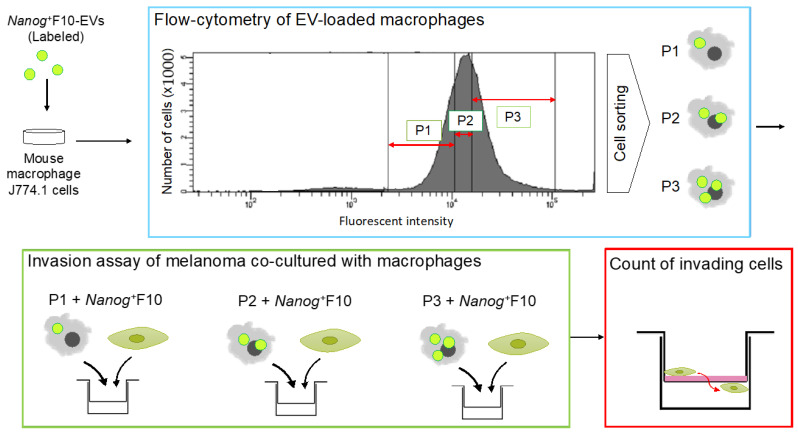
Experimental protocol to analyze the effects of EV-uptake quantity on the invasion ability of macrophages. EVs are labeled with a fluorescent probe. J774.1 cells are fractionated, with a cell sorter, into P1, P2, and P3 fractions, respectively, according to the intensities of fluorescence of EVs. Each fraction of J774.1 cells is co-cultured with *Nanog*^+^F10 cells and tested for invasion ability with Transwell^®^ invasion assay kits. The number of melanoma cells that invade the Transwell membrane are counted.

**Figure 7 cells-11-03881-f007:**
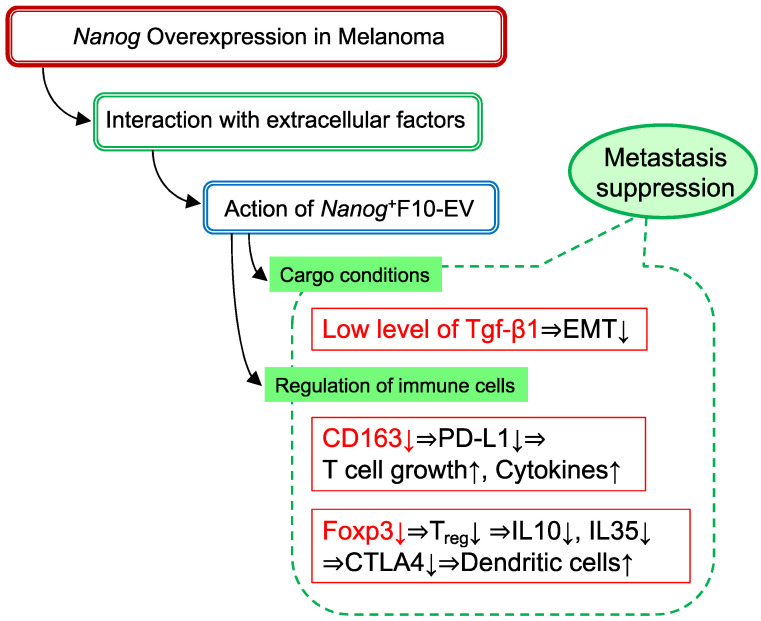
A postulated mechanism of metastasis-suppressive effects by *Nanog*^+^F10-EV.

**Table 1 cells-11-03881-t001:** Effects of the alteration of Nanog expression levels on cancer cell properties.

Type of Cancer [Cell Line]	NEL *^1^ Altering Factor [NEL]	Effects on Cancer Cell Functions	P or S *^2^	Signaling Factors Downstream of Nanog [Role as a Marker of Cell Function]		Ref.
Breast cancer [MCF7]	OE *^3^ [High]	Cell adhesion ↑	P	*Itga9* ↑, *Cldn11* ↑, *Cldn10* ↑, *Cldn6* ↑	Array analysis	[6]
		Ca signaling ↑		*Atp2a1* ↑, *Cacna1h* ↑, *Gna14* ↑, *Cacna1g* ↑		
		Focal adhesion ↑		*Pdgfrα* ↑, *Pdgfrβ* ↑, *Itga11* ↑, *Itgb5* ↑, *Rasgrf1* ↑		
		p53 signaling (tumor suppressor gene) ↓		*Gadd45a* ↓, *Gadd45g* ↓, *Pten* ↓, *Bax* ↓		
		Tumorigenesis by co-expression with Wnt-1				
Breast cancer [T47D, MCF7, MDA-MB-231]	mTOR inhibitors, Chemotherapeutics [Up]	Stem cell-like phenotype ↑	P	*NANOG* ↑, *SNAIL* ↑, *NODAL* ↑		[7]
Cervical carcinoma [CaSki]	OE [High]	Resistance to complement-dependent cytotoxicity ↑	P	CD59 ↑ through promoter occupancy, CD59 binds C8 and C9 and inhibits MAC (membrane attack complex) formation		[8]
Cervical cancer [TC-1 (P0~P3)]	Vaccination-induced cancer evolution (VICE) [Up]	Resistance to cytotoxic T lymphocyte lysis ↑ Stem-like phenotype ↑	P	Stemness markers (*CD133*, *CD44*, *ALDH*) ↑ in TC-1(P3)		[9]
Colon cancer [SW480, SW620, HT29, CT26]	Furin [Down by furin repression]	NANOG ↓, LGR5 ↓, Calcium transportation ↓	S by *NANOG* ↓			[10]
Colon/Colorectal cancer [HCT116, HT29]	OE [High]	Dormancy ↑, Chemoresistance ↑	P	Cell cycle regulating *P21* ↑, *P27* ↑		[11]
Embryonal carcinoma [NT2/D1]	*MIR630* [Down]	Differentiation ↑	S by *NANOG* ↓	*PKC* ⇒ *miR630* ↑ ⇒ *NANOG 3′UTR* ⇒ *NANOG* ↓		[12]
Somatic cancer [HeLa, HCT116]	Nanog-C/CD2 as a Rad51 inhibitor [High]	Inhibition of chemo-/radiotherapy-generated DNA damages of cancer cells mediated by Rad51	S by Nanog-C/CD2	Interaction of C/CD2 domain with Rad51		[13]
Hepatocellular carcinoma (HCC) [MHCC97H, -L, MHCCLM3, etc.]	OE [High]	EMT ↑, Invasion ↑	P	*Nodal* ⇒ *Smad3*		[14]
Hepatocellular carcinoma [CD133+/CD49f+ TICs from HCC of alcoholic patients, HEK239T]	Obesity, Alcohol, Virus [Up]	Self-renewal ↑, Chemoresistance ↑	P	Obesity, Alcohol ⇒ LPS ⇒ *TLR4* ↑ ⇒ *NANOG* ↑ ⇒ *FAO* ↑, *NANOG* ⇒ *OXPHOS* ↓		[15]
Hepatocellular carcinoma [MHCC97-L]	Co-expression of *Oct4* and *Nanog* [High]	Stemness ↑, EMT ↑	P	*Stat3* ↑, *Snail* ↑		[16]
Melanoma (human) [A375]	OE *^3^ [High]	Amoeboid migration ↑	P	*ARHGAP22* ↑, *DOCK10* ↑, *EPHA2* ↑		[17]
Melanoma (mouse) [B16-F10]	Hypoxia [Up]	Spheroid formation ↑, Proliferation ↑	P	*Tgf-β1* ↑		[18]
Melanoma (mouse) [B16-BL6]	OE [High]	Proliferation ↑, Migration ↑, Invasion ↑	P	*Tgf-β1* ↓		[19]
Melanoma (mouse) [B16-F10]	OE [High]	Proliferation ↑, Migration ↑, Invasion ↑	P	*Tgf-β1* ↓		[20]
Ovarian [6 cell lines, Ascites of patients]	*HK2* [Up]	Migration ↑, Invasion ↑, Metastasis ↑	P	*HK2* ⇒ *FAK* ↑ ⇒ *ERK1/2* ↑ ⇒ *MMP9* ↑		[21]
		Stemness ↑	P	*HK2* ⇒ *FAK* ↑ ⇒ *ERK1/2* ↑ ⇒ *NANOG · SOX9* ↑		
Pancreatic cancer stem cell (CSC) [AsPC-1, PANC-1, CSCs from primary tumors]	α-Mangostin [Down]	EMT ↓	S by Nanog ↓	*Gli* signaling (*Nanog*, *Oct4*, *c-Myc*, *Sox-2*, *KLF4*) ↓ [human pancreatic CSCs marker: CD133+ /CD44+/CD24+/ ESA+]		[22]
Prostate cancer [DU145, PC3, 22Rv1]	OE [High]	Escape from NK cell attack ↑	P	ICAM1 ↓		[23]
Prostate cancer [Xenograft models (LAPC4, LAPC9)]	Endogenous NANOG [Varied]	Castration resistance ↓	P	NANOG co-occupies FOXA1 and androgen receptor loci ⇒ pro-differentiation genes ↓		[24]
Esophageal squamous carcinoma [Eca109]	OE [High]	Proliferation ↑, Invasion ↑, Stemness ↑	P	*IL6* ↑, *STAT3* ↑, *CCL5* ↑, *VEGFA* ↑, *CCND*1 ↑, *Bcl-xL* ↑		[25]
Head and neck squamous cell carcinomas (HNSC) [QLL1, SCC15, SCC25]	CD44(+) [Higher than in CD44(-) cells]	Migration ↑, Invasion ↑, Radiotherapy resistance ↑, EMT ↑	P	*NANOG* and *ERK1/2* synergistic effects		[26]
HNSC	Patients [Varied]		S	[NANOG and SOX2 ⇒ Better prognosis]		[27]
Nasopharyngeal Carcinoma	Patients [Varied]		P	High frequency of Nanog, OCT4 at tumor invasive front, lymph node metastasis		[28]
Oral squamous cell carcinoma	Patients with/without lymph node metastasis [Varied]		P	Positive-correlation [*Nanog*] vs [*OCT4*, *NOTCH1*, *AGR2*, *KLF4*] at mRNA [NANOG protein/mRNA:Poor prognosis marker?]		[29]
Cancer stem cell (CSC) models [HNSC, non-small lung cancer, colon cancer, A549]	NANOG overexpressed or varied [High]	Migration ↑, Invasion ↑, EMT ↑	P	*BMI1* ⇒ *E-Cadherin* ↓	[NANOG as a CSC marker (proposed)]	[30]
				*SNAIL2* ⇒ *E-Cadherin* ↓		
				*BMI1* ⇒ *SNAIL1* ⇒ *E-Cadherin* ↓		
				*SNAIL2*, *BMI1* ⇒ *SNAIL1*		
		Proliferation ↑, Self-renewal ↑, Chemoresistance ↑	P	*BMI1*		
		Anti-apoptosis ↑, Chemoresistance ↑	P	*NANOG · STAT3* ⇒ *miR21* ⇒ *PDCD4*(cancer resistance gene) ↓		
	Hedgehog signal (Hh), FAK [Varied]	Stemness ↑	P	Hh ⇒ PATCH1/2 ↓ ⇒ *SMO* ↑ ⇒ *GLI1* ↑ ⇒ *NANOG* ↑ ⇒ *GLI1* ↑ (Positive feedback)		
		Anti-apoptosis ↑	P	*FAK* ⇒ *NANOG* ↑ ⇒ *p53* ↓ ⇒ *NANOG* ↓ (Negative feedback)		
PD-1 therapy treated patients, Anti–PD-1 therapy model mice	Patients with 19 types of cancer [High]	T-cell invasion ↓, Resistance to cytotoxic T-cell ↑	P	*NANOG* ⇒ *HDAC1* ⇒ *Cxcl10* ↓, *MCL1* ↑ [Antitumor immunity marker]		[31]

*^1^ NEL: Nanog expression level; *^2^ P or S: promotive or suppressive; *^3^ OE: overexpression; ↑: up-regulated; ↓: down-regulated.

## Data Availability

Not applicable.

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
