# Peer review of "Novel Roles of Nanog in Cancer Cells and Their Extracellular Vesicles"

_cells, 2022, doi:10.3390/cells11233881_

Round 1
Reviewer 1 Report
This review describes Nanog-dependent phenomena and relevant signaling factors. The author focused on Nanog overexpression in melanoma mouse models and properties of metastasis-suppressive potentials of the extracellular vesicles (EVs) of the melanoma. This review contains interesting information but needs major improvements of the text, figures and tables in the following points.
-Abstract: “EV-based” needs explanation of abbreviation in the sentence.
-Fig. 1 needs legends for explanation. The bottom left part is missing.
-Gene names and genes for mRNA should be in italic.
-Fonts in Table 1 is too small to read. Table 1 and the sections of Results are not well organized and should be divided into two tables of cancer types and cancer conditions/ treatments. Fig. 2 should be also changed accordingly.
-Fig. 3: The names of “7 genes” and “3 genes” should be described in the legends.
-Fig. 5: “N” in the chemical names should be in italic.
-Page 11, line 401: “DAMP” should be corrected to “DAMPs”.
-Page 12, 6.3. : “pg/microg” should be defined.
-Fig. 6 needs legends for the explanation of experiments and citation of the original publication source. The figures of cells and EVs also need explanations.
-Fig. 7 and the legend should be also improved with additional explanations and the citation of the original publication source. The position of “(A)” & “(B)” should be at each top.
-Abbreviations used without explanations should be corrected.
-Page 15, line 525: incomplete.
Author Response
Thank you for your kind and helpful suggestions. We believe the manuscript has been properly revised.
Comments to the author: [1-1]~[1-12]
[1-1] Abstract: “EV-based” needs explanation of abbreviation in the sentence.
[Res.1-1] “EV-based” was revised as “extracellular vesicle (EV)-based”.
[1-2] Fig.1 needs legends for explanation. The bottom left part is missing.
[Res.1-2] Added the following legends for Fig.1.
“Nanog expression level in cancer cells can be changed by genetic, chemical, microenvironmental, or physical factors. The higher the Nanog expression level, the higher the metastatic potential. The role of EVs in cancer metastasis has been thought to follow the metastatic potential of cancer cells. That is, EVs derived from metastatic cancer cells exhibit metastasis-promoting effects. However, in the case of cancer cells with very high metastatic potential, contrary to expectations, EVs may promote cancer metastasis.”
The bottom left part of Fig.1 was not missing. Probably it might be a problem of Figure file conversion. The position of the figure was adjusted.
[1-3] Gene names and genes for mRNA should be in italic.
[Res.1-3] Denoted gene names and mRNA names in italic throughout.
[1-4] Fonts in Table 1 is too small to read. Table 1 and the sections of Results are not well organized and should be divided into two tables of cancer types and cancer conditions/treatments. Fig. 2 should be also changed accordingly.
[Res.1-4] Table 1 was divided into three parts and reorganized.
[1-5] Fig.3 The names of “7 genes” and “3 genes” should be described in the legends.
[Res.1-5] The names of these genes were inserted in the legends for Fig.3.
[1-6] Fig.5 “N” in the chemical names should be in italic.
[Res.1-6] “N”s in the full names of 2NBDG and 2NBDLG in the legends for Fig.5 were made italic.
[1-7] Page 11, line 401: “DAMP” should be corrected to “DAMPs”.
[Res.1-7] “DAMP” was changed to “DAMPs”
[1-8] Page 12, 6.3.: “pg/microg” should be defined.
[Res.1-8] “pg/microg” was changed to “pg/μg EVs”
[1-9] Fig.6 needs legends for the explanation of experiments and citation of the original publication source. The figures of cells and EVs also need explanations.
[Res.1-9] Fig.6 is unpublished data and deleted. Explanation in the text was left.
[1-10] Fig.7 and the legend should be also improved with additional explanations and the citation of the original publication source. The position of “(A)” & “(B)” should be at each top.
[Res.1-10] Fig.7(B) was deleted because of unpublished data. Fig.7(A) was left to help understanding an idea of the test method.
[1-11] Abbreviations used without explanations should be corrected.
[Res.1-11] Explanations were added to the following abbreviations: HNSCC
[1-12] Page 15, line 525 incomplete.
[Res.1-12] “We expect that similar phenomena will be reported with various types of cancer before long.” was revised to “We expect that similar anti-metastasis effects will be observed for EVs derived from Nanog-overexpressing cells of other cancers in the near future.”.

Reviewer 2 Report
This review aimed to summarize the roles of Nanog in cancer cells and their extracellular vesicles (EV). Several points need to be noted to improve the quality of this manuscript as below.
1) As to the novel roles of Nanog their extracellular vesicles in cancers, in fact, the recent advances by the author’s groups about Nanog overexpressing melanoma cells and EV were mainly discussed in this review. This title should be changed, or more studies about Nanog-EV in other cancers should be depicted. For example,
â‘ Zhao LJ, et al. Lysine demethylase LSD1 delivered via small extracellular vesicles promotes gastric cancer cell stemness. EMBO Rep. 2021 Aug 4;22(8):e50922.
â‘ Huang H. RAB27A-dependent release of exosomes by liver cancer stem cells induces Nanog expression in their differentiated progenies and confers regorafenib resistance. J Gastroenterol Hepatol. 2021 Dec;36(12):3429-3437.
â‘¡ Ding C, et al. Warburg effect-promoted exosomal circ_0072083 releasing up-regulates NANGO expression through multiple pathways and enhances temozolomide resistance in glioma.J Exp Clin Cancer Res. 2021 May 11;40(1):164.
2) The text needs to be re-organized. For example, “3.6. Cancer stem cells” should be at the back of “3.12”. In additionally, many images (Figure 3, 6, and 7) about the experimental data were provided in this paper. Are these images had been published or not? If not, it is not appropriate to list here. If they have been published, please make the remark (the Ref.)
3) As to “Cancer metastasis” in the Introduction section, there has been reported that cancer metastasis should be thought as a bidirectional process, and it has been validated in breast cancer, colon cancer and melanoma (â‘ Kim MY, Oskarsson T, Acharyya S, Nguyen DX, Zhang XH, Norton L, Massagué J. Tumor self-seeding by circulating cancer cells.Cell. 2009 Dec 24;139(7):1315-26). An interesting paper recently proposed that nasopharyngeal carcinoma should be the multidirectional process that self-seeding of CTCs or metastatic tumor cells, or self-feeding of their releasing soluble factors such as exosomes, cytokines and chemokines to primary tumor sites. (Nasopharyngeal Carcinoma Ecology Theory: Cancer as Multidimensional Spatiotemporal “Unity of Ecology and Evolution” Pathological Ecosystem. Preprints. 2022; 2022100226. Please check, â‘¡ https://www.preprints.org/manuscript/202210.0226/v1). These views should be complemented in the Introduction section to make it updated.
4) In the Abstract, “Use of EV-based” should be “Use of extracellular vesicles (EV)-based”.
5) In the “3.12 Squamous cell carcinoma”, NANOG also serves as an independent prognostic factor in nasopharyngeal carcinoma (NPC), which is the most frequent head and neck tumor in South China (PLoS One. 2013;8(2):e56324. doi: 10.1371/journal.pone.0056324).
Author Response
Thank you for your kind and helpful suggestions. We believe the manuscript has been properly revised.
Comments to the author: [2-1]~[2-5]
[2-1] As to the novel roles of Nanog their extracellular vesicles in cancers, in fact, the recent advances by the author’s groups about Nanog overexpressing melanoma cells and EV were mainly discussed in this review. This title should be changed, or more studies about Nanog-EV in other cancers should be depicted. For example,
(*1) Zhao LJ, et al. Lysine demethylase LSD1 delivered via small extracellular vesicles promotes gastric cancer cell stemness. EMBO Rep. 2021 Aug 4;22(8):e50922.
(*2) Huang H. RAB27A-dependent release of exosomes by liver cancer stem cells induces Nanog expression in their differentiated progenies and confers regorafenib resistance. J Gastroenterol Hepatol. 2021 Dec;36(12):3429-3437.
(*3) Ding C, et al. Warburg effect-promoted exosomal circ_0072083 releasing up-regulates NANGO expression through multiple pathways and enhances temozolomide resistance in glioma. J Exp Clin Cancer Res. 2021 May 11;40(1):164.
[Res.2-1] Suggested references (*1) and (*2) were added to the revised text section 5.1. Reference (*3) was already cited in the previous text.
[2-2] The text needs to be re-organized. For example, “3.6. Cancer stem cells” should be at the back of “3.12”. In additionally, many images (Figure 3, 6, and 7) about the experimental data were provided in this paper. Are these images had been published or not? If not, it is not appropriate to list here. If they have been published, please make the remark (the Ref.)
[Res.2-2] “3.6. Cancer stem cells” was moved to the back of “3.12”. Fig. 3 was depicted based on the result of reference [19 (previously)]. Fig.6 and Fig.7(B) were deleted because they were unpublished data. Fig.7(A) was left to help understanding an idea of the test method.
[2-3] As to “Cancer metastasis” in the Introduction section, there has been reported that cancer metastasis should be thought as a bidirectional process, and it has been validated in breast cancer, colon cancer and melanoma ((*4) Kim MY, Oskarsson T, Acharyya S, Nguyen DX, Zhang XH, Norton L, Massagué J. Tumor self-seeding by circulating cancer cells. Cell. 2009 Dec 24;139(7):1315-26). An interesting paper recently proposed that nasopharyngeal carcinoma should be the multidirectional process that self-seeding of CTCs or metastatic tumor cells, or self-feeding of their releasing soluble factors such as exosomes, cytokines and chemokines to primary tumor sites. ((*5) Nasopharyngeal Carcinoma Ecology Theory: Cancer as Multidimensional Spatiotemporal “Unity of Ecology and Evolution” Pathological Ecosystem. Preprints. 2022; 2022100226. Please check, https://www.preprints.org/manuscript/202210.0226/v1). These views should be complemented in the Introduction section to make it updated.
[Res.2-3] A description about the multidirectional process of CTCs was added to Introduction by citing references (*4) (Please let us skip citing the reference *5 because of “not pere revied document).
[2-4] In the Abstract, “Use of EV-based” should be “Use of extracellular vesicles (EV)-based”.
[Res.2-4] Corrected as advised.
[2-5] In the “3.12 Squamous cell carcinoma”, NANOG also serves as an independent prognostic factor in nasopharyngeal carcinoma (NPC), which is the most frequent head and neck tumor in South China ((*6) PLoS One. 2013;8(2):e56324. doi: 10.1371/journal.pone.0056324).
[Res.2-5] Suggested reference (*6 ) was cited and explanation was added to “3.12”

Round 2
Reviewer 2 Report
No other questions